# Perinatal Exposure to Triclosan Results in Abnormal Brain Development and Behavior in Mice

**DOI:** 10.3390/ijms21114009

**Published:** 2020-06-03

**Authors:** Dinh Nam Tran, Eui-Man Jung, Yeong-Min Yoo, Jae-Hwan Lee, Eui-Bae Jeung

**Affiliations:** Laboratory of Veterinary Biochemistry and Molecular Biology, Veterinary Medical Center and College of Veterinary Medicine, Chungbuk National University, Cheongju, Chungbuk 28644, Korea; mr.tran90tb@gmail.com (D.N.T.); jemman@hanmail.net (E.-M.J.); yyeongm@hanmail.net (Y.-M.Y.); phantom4015@nate.com (J.-H.L.)

**Keywords:** triclosan, brain development, behavioral disorders

## Abstract

Triclosan (TCS) is one of the most common endocrine-disrupting chemicals (EDCs) present in household and personal wash products. Recently, concerns have been raised about the association between abnormal behavior in children and exposure to EDC during gestation. We hypothesized that exposure to TCS during gestation could affect brain development. Cortical neurons of mice were exposed in vitro to TCS. In addition, we examined in vivo whether maternal TCS administration can affect neurobehavioral development in the offspring generation. We determined that TCS can impair dendrite and axon growth by reducing average length and numbers of axons and dendrites. Additionally, TCS inhibited the proliferation of and promoted apoptosis in neuronal progenitor cells. Detailed behavioral analyses showed impaired acquisition of spatial learning and reference memory in offspring derived from dams exposed to TCS. The TCS-treated groups also showed cognition dysfunction and impairments in sociability and social novelty preference. Furthermore, TCS-treated groups exhibited increased anxiety-like behavior, but there was no significant change in depression-like behaviors. In addition, TCS-treated groups exhibited deficits in nesting behavior. Taken together, our results indicate that perinatal exposure to TCS induces neurodevelopment disorder, resulting in abnormal social behaviors, cognitive impairment, and deficits in spatial learning and memory in offspring.

## 1. Introduction

Triclosan (TCS) is widely used in consumer products as an antibacterial agent and is found in toothpaste, mouthwash, soap, deodorants, textiles, toys, and medical devices [1]. TCS is a suspected endocrine-disrupting chemical (EDC) with estrogenic/androgenic and thyroid hormone activities [2,3]. TCS easily passes through biological barriers and accumulates in living organisms in fat, liver, brain, blood, and breast milk [4]. In the United States, TCS has been detected in approximately 75% of adults and 85% of pregnant women [5]. In addition, TCS may affect metabolism in mesenchymal stem cells by inhibiting adipocyte differentiation at low concentrations that are not cytotoxic [6]. In an examination of National Health and Nutrition Examination Surveys data, TCS-exposure was associated with an increased body mass index [7]. Moreover, TCS can cross the placental barrier, leading to fetal TCS exposure. Prenatal TCS exposure is reported to affect reproductive hormones mediated by steroidogenic enzymes, and male infants were more vulnerable to such effects [8]. Measurement of urinary TCS concentrations in pregnant women has shown an association between TCS and decreases in birth weight, birth length, head circumference, and gestational age [9]. In rats, disrupted fetal development and embryonic developmental delay can be caused by prenatal exposure to TCS [10]. In mice, exposure to TCS during gestation may impair placental development and nutrient transfer, leading to decreases in fetal body weight by inducing hypothyroxinemia through a reduction in Akt-mTOR signaling [11]. Recently, TCS has been reported to act as a neurotoxicity agent [12]. In a previous study, TCS could impair hippocampal synaptic plasticity and lead to memory impairment in male rats [13]. In adult mice, gavage exposure to TCS at a high dose changed locomotor activity and coordination and induced anxiety behavior [14]. Moreover, TCS may adversely affect neural stem cell viability and survival, and TCS may induce neurotoxicity in developing rat brain [15]. TCS can also evoke neurotoxicity by inducing DNA fragmentation and apoptosis in primary cultures of neocortical neurons through caspase-3, caspase-8, and Fas receptor signal activation [16]. In addition, TCS at a high concentration can slightly delay the development of secondary motor neurons in zebrafish [17]. These findings show that TCS may induce adverse effects on the central nervous system (CNS) functions via apoptosis and oxidative stress [12]. Also, a short-term infusion of TCS, either in the pregnant ewe or in the ovine fetus, has induced slight changes in fetal hypothalamic transcriptome [18]. However, there is limited evidence regarding the effects of multiple exposures to TCS at an early age, thereby restricting current knowledge of the potential adverse effects on the developing brain. Thus, there are still many unresolved questions concerning possible health consequences of TCS intake. In particular, the effects of maternal TCS exposure as well as other EDCs on brain development and behavioral abilities in offspring remain insufficiently described.

Notably, adversity during early-life brain development is associated with an increased risk for the development of behavioral and psychiatric disorders, including depression, anxiety, and schizophrenia [19]. In addition, small disturbances during the early brain development period result in adverse neurodevelopmental outcomes and lead to subsequent disorganization and, eventually, dysfunction of the CNS. Recent studies have shown thyroid hormones to be critical for the development and function of the CNS. Thyroid hormones have been reported to regulate the development and differentiation of neurons and neuroglia [20]. Impairment of thyroid hormone supply may result in severe and irreversible changes to brain development and function, leading to various neurological dysfunctions [21]. In addition, maternal exposure to TCS has been shown to impair to thyroid homeostasis in rat offspring [10]. We designed the present study to identify those aspects of TCS developmental exposure that produce negative neuropathological and functional outcomes. In vitro, we examined the delays on the development of primary cortical neurons by TCS. In addition, we investigated the adverse effects of TCS on the developing brain by treating maternal mice with subcutaneous TCS doses from E9.5 to postnatal day (PND) 28. A comparison of the effects on offspring mice of TCS-treated dams with the effects on behavioral functions was undertaken. The results showed that the TCS-treated offspring showed cognitive dysfunction and anxiety-like behavior. Overall, our results indicate that maternal TCS exposure impairs brain development and produces behavioral changes in offspring.

## 2. Results

### 2.1. TCS Impairs to Growth and Development of Primary Cortical Neuron

A previous study suggested that exposure to TCS delays the development of motor neurons [17]. As TCS can have undesirable effects on neuronal morphology, we first examined the effects of TCS on dendritic and axonal growth of cultured neurons. Primary cortical neurons were collected at E15 and exposed to TCS. We assessed dendrite length and numbers of primary and secondary dendrites at DIV4 (Figure 1A). At the TCS low dose (10^−8^ M), TCS showed no effect on the number of primary dendrites compared to that in vehicle group (Figure 1B). The number of secondary dendrites in TCS 10^−8^ M group were no significant decreased compared to the vehicle group (Figure 1C). However, the length of dendrites was significantly decreased in the TCS 10^−8^ M group compared with the vehicle group (F_2,440_ = 43.93, *p* < 0.0001) (Figure 1D).

At the TCS high dose (10^−6^ M), the length of primary dendrites in the TCS 10^−6^ M group was significantly decreased compared to that in the vehicle group (Figure 1D). The numbers of primary dendrites in the TCS 10^−6^ M group were significantly decreased compared to that in the vehicle group (F_2,150_ = 16.69, *p* < 0.0001) (Figure 1B). The numbers of secondary dendrites were also slightly lower in the TCS 10^−6^ M group compared to the vehicle group (F_2,147_ = 3.594, *p* = 0.0300) (Figure 1C). In addition, compared to the TCS 10^−8^ M group, the TCS 10^−6^ M group exhibited significant decreases in the number of primary dendrites and the length of primary dendrites. There was no significant difference in the number of secondary dendrites between the TCS 10^−8^ M and TCS 10^−6^ M groups.

Regarding axon formation, the numbers of primary-, secondary- axons and the length of axons in the TCS 10^−8^ M group were no significant differences compared to those in the vehicle group (Figure 1E,F). At the TCS high dose (10^−6^ M), the length of axons in the TCS-treated group was significantly decreased compared to that in the vehicle group (F_2,150_ = 26.91, *p* < 0.0001) (Figure 1C).

In addition, the number of primary axons was also smaller in the TCS 10^−6^ M group compared to that in the vehicle group (F_2150_ = 7.594, *p* = 0.0007) (Figure 1F). However, the TCS 10^−6^ M group exhibited no significant difference in the number of secondary axons compare to the vehicle group (F_2,150_ = 0.8790, *p* = 0.4174) (Figure 1G). Compared to the TCS 10^−8^ M group, TCS 10^−6^ M showed significantly decreases in the number of primary axons and the length of primary axons. These results indicate that TCS may impair neurite growth of neurons at an early stage.

### 2.2. TCS Impairs Neuronal Progenitor Cell Proliferation and Cell Death

We next sought to determine the effect of TCS on the proliferation of neuronal progenitor cells. Neuronal progenitor cell proliferation was assessed by applying immunofluorescence with the proliferation marker BrdU on primary cortical neuron cells. The percentage of BrdU^+^ /DAPI cell was quantified. The percentage of BrdU^+^ cells in both TCS 10^−8^ M and TCS 10^−6^ M groups were lower than in the vehicle group (F_2,132_ = 23.78, *p* < 0.0001) (Figure 1H,I). Moreover, there was significant lower the percentage of BrdU^+^ cells in the TCS 10^−6^ M group than that in the TCS 10^−8^ M group (Figure 1H,I).

To further elucidate the effects of TCS on neuronal progenitor cells, we assessed the changes in apoptotic cell abundance by using cleaved caspase-3 staining on primary cortical neuron cells. The percentage of caspase-3^+^/DAPI cell was quantified. As expected, the percentages of caspase-3^+^ cells were markedly higher in the TCS 10^−6^ M group compared to the vehicle group (F_2,150_ = 18.80, *p* = 0.0005) (Figure 1J,K). Additionally, relative to the vehicle group, there were significantly differences in the percentages of cleaved caspase-3^+^ cells in the TCS 10^−8^ M and vehicle groups (Figure 1K). In addition, compared to the TCS 10^−8^ M treatment group, the percentage of cleaved caspase-3^+^ cells in the TCS 10^−6^ M was markedly higher.

Next, we examined the effect of perinatal exposure to TCS on the proliferation of precursor neuronal cells during embryogenesis. The number of proliferating cells in the dentate gyrus (DG) area of TCS-treated and vehicle embryos by injecting pregnant females with BrdU at E18.5 and harvesting embryos 2 h later (Figure 2A). The number of BrdU^+^, Ki67^+^ and *p*-Histone-H3^+^ cells in the DG were determined. The number of BrdU^+^ cells was slightly decreased in the E18.5 TCS-treated groups compared to that of the vehicle group (F_2,15_ = 20.39, *p* < 0.0001) (Figure 2A,B). In addition, the number of cells expressing the mitotic marker Ki67^+^ at DG was significantly lower in TCS-treated groups than in the vehicle group (F_2,15_ = 6.022, *p* = 0.0120) (Figure 2A,C). Moreover, the numbers of cells expressing the mitotic marker phospho-histone H3 (*p*-Histone-H3) at the DG were lower in the TCS-treated groups than in the vehicle group (F_2,15_ = 23.88, *p* < 0.0001) (Figure 2A,D). There was no difference in the number of BrdU^+^, Ki67^+^ and *p*-Histone-H3^+^ cells between the TCS 10 and TCS 100 mg/kg groups (Figure 2A,D). 

Furthermore, there was a significantly lower proportion of cell cycle re-entry in the TCS-treated groups compared to the vehicle (BrdU^+^Ki67^+^/BrdU^+^) (Figure 2E,F) (F_2,15_ = 6.485, *p* = 0.0093). However, TCS 10 mg/kg group showed no significant difference in the proportion of cell cycle re-entry compared to the TCS 100 mg/kg group (Figure 2E,F). These results suggest that TCS can inhibit proliferation and promote apoptosis of neuronal progenitor cells during brain development.

### 2.3. TCS-Exposure Result in Growth and Developmental Delay

To identify any gross developmental delay induced by TCS exposure that might underlie long-term alterations in behavioral functions, we inspected the offspring of TCS-treated dams weekly for any signs of gross physical abnormality and monitored body weight. The TCS 100 mg/kg group exhibited significantly lower birth weight compared to the vehicle and the TCS 10 mg/kg groups at PND 1 (F_2,45_ = 10.34, *p* = 0.0002) (Appendix A). Moreover, the numbers of pups per litter were significantly decreased in TCS-treated groups compared to the vehicle groups (F_2,12_ = 13.73, *p* = 0.0008) (Appendix A). There was no difference in body weight between the TCS 10 mg/kg and vehicle groups at PND1 to PND 119. We did not observe physical anomalies in the offspring of TCS-treated dams, but analysis of pre-weaning and during life-time body weights showed the offspring mice were affected by perinatal exposure to the high dose of TCS (100 mg/kg) (Appendix A). Moreover, the brain weight of the TCS 100 mg/kg group was significantly lower (about 6%) than that of the vehicle group at PND 119 (F_2,20_ = 3.596, *p* = 0.0463) ( Appendix A). Additionally, the ratios of brain weight to whole body weight were markedly decreased in both TCS 10 and TCS 100 mg/kg groups compared to vehicle group (F_2,20_ = 6.609, *p* = 0.0075) (Appendix A). These results suggest that perinatal exposure to TCS impairs brain growth.

Next, we examined whether TCS induced the apoptosis signaling in the brain of adult offspring mice. The levels of cleaved caspasae-3, the proteolytic product of the apoptotic executioner caspase-3, were measured by western blotting in all groups at PND 119 (Appendix A). TCS-treated groups showed markedly higher cleaved caspase-3 levels compared with vehicle group (F_2,9_ = 10.84, *p* = 0.0040) (Appendix A). Additionally, the protein levels of anti-apoptotic Bcl2 were significantly lower in the TCS-treated groups compared to the vehicle group (F_2,9_ = 9.321, *p* = 0.0106) (Appendix A). In addition, compared to TCS 10 mg/kg group, TCS 100 mg/kg group showed no significant differences in the protein levels of cleaved caspase-3 and Bcl2. These results indicate that maternal exposure to TCS treatment during pregnancy and lactation elevated apoptosis in the brain of adult offspring mice.

Additionally, TCS has been reported to have a similar structure to thyroid hormone [22], suggesting it may disrupt the thyroid function. In present study, we found that maternal exposure to TCS during pregnancy and lactation decreased mRNA levels of thyroid hormone-related genes such as thyroglobulin (*Tg*), thyroperoxidase (*Tpo*), sodium-iodide symporter (*Nis*)*,* monocarboxylate transporter 8 (*Mct8*) and thyrotropin receptor (*Tshr*) in the thyroid gland at PND 119 (Appendix A). Conversely, TCS-treated groups showed higher mRNA levels of thyroid hormone receptor alpha (*Trhra*), thyrotropin releasing hormone (*Trh*) in brain and thyrotropin releasing hormone receptor (*Trhr*) in pituitary gland (Appendix A). Moreover, mRNA level of thyroid hormone transport gene type 2 iodothyronine deiodinase (*Dio2*) in brain was markedly higher in TCS-treated groups compared to vehicle group (Appendix A). Meanwhile, the mRNA level of *Mct8* and organic anion transporter family member 1C1 (*Oatp1c1*) were lower in brain of TCS-treated groups compared to vehicle group (Appendix A). These results suggest that the exposure of mouse dams to TCS altered thyroid function and may induce the hypothyroidism in their offspring in adult life.

### 2.4. TCS-Exposure Induce Cognition Dysfunction in Offspring Mice

To address the functional relevance of the morphological and biochemical changes observed in the TCS-treated groups, we examined the cognitive, social, and emotional behaviors in the offspring mice. The postpartum female mice exposed to TCS displayed normal maternal behavior, including pup retrieval, licking of pups, nest building, crouching over grouped pups in a well-defined nest, maternal aggression and milk spot in stomach of offspring mice (data not shown). We first performed the Morris water maze test to assess the spatial learning and memory impairment of members of the TCS-treated groups. Compared to the vehicle group, both TCS-treated groups exhibited increased time to find the hidden platform during the acquisition phase (four trials per day for four successive days) (Figure 3A). On test days 5 to 9, the TCS-treated groups improved the time to find the platform, but the time was still markedly longer compared to the vehicle group (Figure 3A). In addition, the TCS 10 mg/kg and TCS 100 mg/kg groups exhibited no significant differences in the time to find the platform (Figure 3A). This indicate that the capacity for spatial learning in TCS-treated groups is delayed. In the probe test, the platform was removed from the pool after completion of a training phase. The vehicle group displayed a significantly higher in the number of target platform crossings compared to those of the TCS 10 and TCS 100 mg/kg (F_2,125_ = 4.807, *p* = 0.0097) groups (Figure 3C). In addition, the vehicle group spent more time in the area in which the platform was originally located when compared to the TCS 10 mg/kg and TCS 100 mg/kg (F_2,125_ = 5.159, *p* = 0.0054) groups (Figure 3D). Visual representations recorded for all swim tracks of each group showed that the TCS-treated groups exhibited lower proximities to the old platform quadrant than in the vehicle group (Figure 3B). But, the vehicle and TCS-treated groups did not show significant differences in swimming distances (F_2,125_ = 0.8973, *p* = 0.0321) (Figure 3E). Additionally, the TCS 10 mg/kg and TCS 100 mg/kg groups exhibited no significant differences in swimming speeds compared to vehicle group (F_2,125_ = 1.068, *p* = 0.0259) (Figure 3F).

We also assessed recognition memory by using a novel-object recognition test. The mice were explored to two identical objects (similar shape and color), and after 6 h, one of the objects was replaced by a new novel object (different shape and color from that of the old object) (Figure 3G).

With regard to the time spent on each object, we found that the vehicle group spent more time approaching and in proximity to the novel object (vehicle; Familiar = 45.97 ± 1.71, vehicle; Novel = 54.03 ± 1.71, t = 3,339, *p* = 0.0021) (Figure 3H). In contrast, both the TCS 10 mg/kg and TCS 100 mg/kg groups showed no preference for exploring either the familiar or the novel object (TCS 10 mg/kg; Familiar = 48.00 ± 1.62, TCS 10 mg/kg; Novel = 52.00 ± 1.62, (t = 3.041, *p* = 0.0545) and TCS 100 mg/kg; Familiar = 49.39 ± 1.82, TCS 100 mg/kg; Novel = 50.61 ± 1.82, (t = 0.5095, *p* = 0.6144)) (Figure 3H). Furthermore, no significant gender differences were found (data not shown). Taken together, our results suggest that members of the TCS-treated group have impaired cognitive functioning in behavioral areas such as spatial and nonspatial learning and memory.

### 2.5. TCS Induces Social Deficiency in Mice

Next, we used the three-chamber social test to assess social interaction and discrimination of social novelty for each mouse. In addition, analysis of the preference index for social behaviors allows the direct comparison of the social behaviors of the groups. In the sociability test, the mice explored the three-chamber apparatus with the presence of an unfamiliar mouse in one of the side-chambers (Figure 4A). 

Both the vehicle and TCS-treated groups spent significantly more time in the chamber containing the unfamiliar mouse (S1) than in the empty chamber (E) (vehicle; Empty = 153.83 ± 5.59, Stranger I = 274.15 ± 8.67, (t = 10.40, *p* < 0.0001); TCS 10 mg/kg; Empty = 186.19 ± 11.16, Stranger I = 237.18 ± 16.23, (t = 2.589, *p* = 0.0175); TCS 100 mg/kg; Empty = 151.51 ± 7.40, Stranger I = 251.19 ± 13.90, (t = 7.021, *p* < 0.0001)) (Figure 4B). Analysis of the preference index for social interaction showed significant index reductions in both the TCS 10 and TCS 100 mg/kg groups compared to that in the vehicle group (F_2,28_ = 17.76, *p* < 0.0001) (Figure 4C). In the social novelty test, a second unfamiliar mouse (S2) was introduced instead of an empty cage. In this case, the vehicle group showed a greater preference for novel chamber S2 than for familiar chamber S1 (vehicle; Stranger I = 142.92 ± 14.48, Stranger II = 220.07 ± 15.36, (t = 2.815, *p* = 0.0092)) (Figure 4A,D). Unlike the vehicle group, the TCS-treated groups demonstrated no preference for the S2 stranger mouse over the S1 familiar mouse (TCS 10 mg/kg; Stranger I = 207.47 ± 11.56, Stranger II = 192.23 ± 19.74, (t = 0.6733, *p* = 0.5805); TCS 100 mg/kg; Stranger I = 183.92 ± 6.94, Stranger II = 212.21 ± 13.12, (t = 1.997, *p* = 0.0549)) (Figure 4D). Compared to the vehicle group, there were markedly lower preference indices for the novel stimulus in the TCS 10 and TCS 100 mg/kg groups (F_2,28_ = 8.327, *p* = 0.0015) (Figure 4E). Interestingly, relative to the TCS 100 mg/kg group, the TCS 10 mg/kg group had markedly lower preference indices for social interaction (Figure 4E). These results demonstrate that perinatal exposure to TCS impaired social behaviors in offspring mice.

### 2.6. TCS-Exposure Induces Anxiety-Like Behavior, but does not Affect Depression-Like Behavior in Mice

The open-field test was used to analyze motor activity (reflected by the distance traveled) and anxiety-like behavior (reflected by the time spent in the center zone). The TCS 100 mg/kg group spent significantly less time in the open field center zone compared to the vehicle group (F_2,43_ = 3.962, *p* = 0.0245) (Figure 5B). 

However, the TCS 10 mg/kg group showed no difference in time spent in the open field center compared to that of the vehicle and TCS 100 mg/kg groups (Figure 5B). Additionally, compared to the vehicle group, there were significantly fewer open field center entries in the TCS 100 mg/kg group but no difference in entries in the TCS 10 mg/kg group (F_2,43_ = 13.75, *p* < 0.0001) (Figure 5C). Moreover, the TCS 100 mg/kg group displayed significantly lower open field center entries compared to those of the TCS 10 mg/kg group (Figure 5C). However, the number of entries into the center zone in the TCS 10 mg/kg and vehicle groups were not significantly different (Figure 5C). There were no differences in distance traveled (Figure 5D) between the vehicle and TCS 10 mg/kg or TCS 100 mg/kg groups (F_2,43_ = 1.317, *p* = 0.4436). In addition, the TCS-treated groups showed no difference in velocity compared to vehicle group (F_2,43_ = 1.275, *p* = 0.4637) (Figure 5E). There was also no significant difference in velocity and distance traveled between the TCS 10 mg/kg and TCS 100 mg/kg groups (Figure 5D,E). Taken together, the results indicate that perinatal exposure to TCS elevates anxiety-like behavior in offspring mice.

Next, we examined the potential of TCS to induce depression by assessing depressive-like behavior based on using tail suspension and forced swimming tests. The results showed that the immobility times of TCS-treated groups were not significantly different from the vehicle group in the tail suspension test (F_2,34_ = 0.4342, *p* = 0.5535) (Figure 5F). There was also no significant difference in the immobility times of the TCS 10 mg/kg and TCS 100 mg/kg groups (Figure 5F). The forced swimming test results revealed no significant differences between the TCS-treated groups and the vehicle group (F_2,34_ = 0.02233, *p* = 0.0522) (Figure 5G). These results show that perinatal exposure to TCS does not affect the exhibition of depression-like behavior in offspring mice.

### 2.7. TCS-Exposure Reduces Nesting-Behavior in Mice

Nesting behavior is a motivated, goal-directed activity requiring orofacial and forelimb dexterity. To quantify the nesting behavior of offspring mice at six-weeks of age, we used Deacon’s five-point rating system. Vehicle-treated mice-built nests perfectly and within 12 h, leaving no unshredded pieces of nestlet material (Appendix A). In contrast, both the TCS 10 mg/kg and TCS 100 mg/kg groups failed to construct nests, achieving low nesting scores compare to vehicle (F_2,33_ = 9.665, *p* = 0.0005) (Appendix A). These results indicate that there was severe impairment of nesting behavior in offspring mice of TCS-exposed dams.

## 3. Discussion

In recent years, there have been increasing concerns about the effect of EDCs on human health. Of special concern is the placental transfer of EDCs from mother to fetus; thus, fetuses and children are particularly vulnerable to the effects of EDCs [23,24]. Unsurprisingly, EDCs have been shown to be involved in many neurological disorders. Environmental factors, such as exposures to polychlorinated biphenyls (PCBs), dichlorodiphenyltrichloroethane (DDT)-, bisphenol A (BPA)- and phthalates have been implicated in the pathogenesis of neurodevelopmental disorders, including autism spectrum disorder (ASD) [25]. Exposure to BPA impairs spatial memory in postnatal male mice, as well as mouse motor activity, anxiety-like behavior and social behaviors. Meanwhile, exposure to PCBs not only produced hyperactivity and impulsiveness, but it also altered social behaviors in rats. In humans, maternal exposure to several EDCs has been associated with adverse neurodevelopmental performances among the children aged 1–2 years [26]. In addition, exposure to EDCs may disrupt brain development [25]. Many studies have reported the adverse effects of EDC on mental health, but the effects of EDC exposure at an early age have not yet been fully elucidated. Herein, we investigate the effects of TCS, a common EDC around the world, on behaviors and on the development of the brain in a perinatal exposed mouse model. For this study, we examined social interaction, cognition, and anxiety performance levels.

TCS has been identified in the milk and plasma of nursing mothers in Sweden and Australia [27,28]. TCS was also found in urine and cord blood samples obtained from 181 expectant mothers in the United States [29]. Furthermore, TCS was shown to impair the viability and survival of neuronal cells by inducing DNA fragmentation and apoptosis [16]. In cultured rat neural stem cells, TCS produced neurotoxicity in developing rat brains [15]. In another study, TCS slightly delayed the development of secondary motor neurons in zebrafish [17]. Thus, we hypothesized that TCS may disrupt the formation of dendrites and axons. Indeed, we observed that TCS impairs axon formation by reducing the numbers of primary and secondary axons as well as dendrites on DIV4 of primary cortical neuron culture. Moreover, TCS can affect axon elongation by decreasing axon length. In addition, dendrite length was also decreased by TCS. These results suggest that TCS impairs the formation and elongation of axons and dendrites in vitro. In the present study, we also found decreased proliferation and enhanced apoptosis of TCS-treated neuronal progenitor cells. Recent studies have demonstrated that all neuron and glial cells are derived from embryonic and postnatal neural stem cells and neuronal progenitor cells [30]. In addition, the neurogenic and gliogenic potentials of progenitor cells are mainly controlled during embryogenesis [31]. Additionally, alteration in dendrite morphology and defects in neuronal development, including changes in dendrite branching patterns, fragmentation of dendrites, and retraction or loss of dendrite branching, contribute to several neurological and neurodevelopmental disorders, such as autism spectrum disorders, Alzheimer’s disease, schizophrenia, Down syndrome, Fragile X syndrome, Rett syndrome, anxiety, and depression [32]. These findings suggest that TCS may impair neurogenesis and gliogenesis during embryogenesis in mice.

In addition, as an EDC, TCS has endocrine-disrupting properties and may affect the regulation of estrogen, testosterone, and thyroid hormone levels in animal experiments. Some studies have shown that exposure to TCS reduces the level of thyroid hormone in rats [10], while other studies have demonstrated that TCS exhibits estrogenic and androgenic activities [2]. In an earlier study, estrogen was reported to be involved in the modulation of neuronal differentiation, notably by influencing cell migration, cell survival and death, and synaptic plasticity of neurons [33]. Moreover, thyroid hormone balance during pregnancy is important for the neurodevelopment of fetus [34]. In early pregnancy, low levels of thyroxine can lead to neurological disabilities and underdevelopment of the cortex [35]. The effects of TCS on human health remains incompletely elucidated because of the lack of clarity on its hormonal activity.

Both human and mouse brain development are remarkably complex processes and begin during early gestation and continue into the postnatal period. Thus, in this study, TCS was introduced on E9.5 in mice to investigate its effects on brain development. At high dose, TCS showed a decrease in birth weight and the numbers of offspring mice compared to vehicle group. This suggests that TCS can have adverse effects on fetal development [9,10,11]. The body weights of offspring mice were consistently lower than that of the vehicle group throughout the lifetime of the mice. Interestingly, the brain weights of adult offspring mice were also significantly lower in the TCS 100 mg/kg group compared to that of the vehicle group, indicating that exposure to a high dose of TCS at an early gestational stage may impair brain development. This may be the result of TCS impairing the early-stage development of neurons, with such changes persisting into postnatal life, resulting in adverse consequences in adulthood. Indeed, in the present study, TCS reduced proliferation of neuronal progenitor cells during brain development. Moreover, TCS promoted apoptosis in the brain of adult offspring mice by increasing the expression of cleaved caspase-3. In addition, anti-apoptotic Bcl2 protein levels were decreased by perinatal exposure to TCS. The Bcl2 family proteins have been reported to activate caspase through the intrinsic and extrinsic apoptotic pathway in a model of neuronal apoptosis [36]. In addition, Bcl2 family proteins play an important role in initiating and inhibiting apoptosis during neuronal development and injury [37]. The downregulation of Bcl2 gene expression levels enhanced oxidative stress and altered in antioxidants in the brain [38]. Our results suggest that perinatal exposure to TCS promotes apoptosis in the adult brain by regulating the expression level of Bcl2. In accordance, TCS treatment showed a decrease in the brain weight of offspring mice. Future work will focus on the cell death pathways induced by the TCS and which region be affected in the brain.

In an early study, thyroid hormone was shown to promote Bcl-2 expression and prevent apoptosis of early differentiating cerebellar granule neurons [39]. Thyroid hormone deficiency decreased the expression of Bcl2 and led to enhanced apoptosis in the developing cerebellum [40]. Additionally, TCS has a similar structure to the thyroid hormones [22] indicates it can have adverse effects on the thyroid system. Indeed, TCS has been showed to alter serum thyroid hormones levels including triiodothyronine (T3) and thyroxine (T4) [41]. Analysis of the thyroid hormone synthesis, TCS has been reported to alter the expression levels of thyroid hormone responsive genes in the hypothalamus-pituitary-thyroid axis [42]. In the present study, perinatal exposure to the TCS reduction the expression of *Pax8/Nkx2-1* associated with the reduction of their transcriptional activity (*Tshr*, *Tg*, *Tpo, Nis* and *Mct8*) which involved in the thyroid hormone synthesis and secretion in adult offspring mice. Importantly, the increased expression of *Trh* and *Trhr* indicated the reduced action of thyroid hormone in the hypothalamus-pituitary axis. In fact, *Trh* and *Trhr* mRNA are known to be down-regulated by T3 [43]. Moreover, the *Nis* dysfunction impaired to the thyroid hormonogenesis in human and caused congenital goitrous hypothyroidism [44]. In addition, thyroid hormone, T4 and T3, play important roles in all the events of brain development [45]. In brain neurons, T4 and T3 are facilitated by specific thyroid hormone transporters, mainly *Mct8* and *Oatp1c1*. Recently, *Mct8* and *Oatp1c1* double knockout mice showed hypothyroidism. Additionally, expression of *Dio2*, that is known to be regulated by T3 and T4 serum levels, was markedly increased in the brain of *Mct8/Oapt1c1* double knock-out mice [46]. Furthermore, the hypothyroid condition was confirmed by the reduction of *Pax8/Nkx2.1* expression associated with their reduced transcriptional activity might contribute to the decreased expression of genes/proteins involved in the TH synthesis and secretion (*Tshr*, *Nis*, *Tpo*, *Tg* and *Mct8*) [47]. Our data suggests that perinatal exposure to TCS resulted in the inactivation of thyroid synthesis, although the thyroid hormone levels were not measured. In addition, thyroid hormone deficiency influences the development of axon and dendrites process, by which results in the morphological alteration during the early stage of brain development [48]. Moreover, hypothyroidism has been reported to associate with bipolar affective disorders, depression, or loss of cognitive functions [49].

To address the functional relevance of the morphological and biochemical changes in the TCS-treated groups, we examined cognitive, social, and emotional behaviors. In previous studies, TCS has been reported to impair hippocampal synaptic plasticity [13] and fetal hypothalamic transcriptome [18], which may be reflected by the observed spatial learning and memory deficits and cognitive impairment in the present study. In addition, perinatal exposure to TCS impaired to brain development at an early stage may lead to abnormal social behavior and anxiety-behavior in adulthood.

In conclusion, we have observed that exposure to TCS during the perinatal period disrupted mouse brain development by affecting neurogenesis and neurite growth. Consequently, the offspring mice from the TCS-treated dams showed impairments to memory, social behavior, and anxiety behavior. However, as an EDC, TCS may affect thyroid activities [2,10]. Identification of the specific pathways associated with the TCS effects may help ameliorate the potential acute cell loss and long-term functional disruptions induced by exposure of the developing brain to TCS. More research is needed to further clarify the mechanisms behind the effects of TCS on brain development and subsequent behavior.

## 4. Materials and Methods

### 4.1. Animals and Treatments

Specific pathogen-free adults C57BL/6J male and female mice (8-weeks-old, 25–30 g) were obtained from Samtaco (Osan, Gyeonggi, Republic of Korea). The mice were housed in polycarbonate cages under a controlled environment conditions as in previous work [24]. After the acclimatization period, female mice were mated with adult male mice overnight at a proportion of 2:1, and the presence of a vaginal plug was set as embryonic day (E) 0.5. The maternal mice were randomly divided into three groups including vehicle group and two exposure groups (*n* = 5 mouse/group, each mouse was maintained in each cage) and they chronically received different TCS levels by subcutaneous injection. Mice in the three groups were given 0.2 mL of corn oil with TCS concentrations of 0, 10 and 100 mg/kg every day from E9.5 to postnatal day (PND) 28.5. In mice, the dose of 10 mg/kg TCS exposure caused an equivalent urinary TCS level as those in the high-TCS abortion patients [50]. In order to more clarify the effect of TCS on the brain development, high dose of TCS (100 mg/kg) was used. After weaning (PND 28.5), the female and male offspring were separated and housed in group of 3–5 animals. All experimental protocols were approved by the Institutional Animal Care and Use Committee of Chungbuk National University (IACUC) and all experiments were carried out in accordance with the relevant guidelines and regulations.

### 4.2. Primary Cortical Neuron Culture

Primary cortical neuron culture was performed as described previously [51]. The primary cortical brains were collected from E15.5 mouse embryos and dissociated to single cells after digestion with trypsin (Celgene, Summit, NJ, USA). 1x10^5^ neuronal cells were plated in poly-D-lysine coated 24-well plates and cultured in Neurobasal medium/DMEM (1:1) with B27 supplement, penicillin, streptomycin and glutamine (Gibco-BRL, Gaithersburg, MD, USA) at 37 °C in a humidified incubator with 5% CO_2_/95% air. The day of plating was considered day in vitro 0 (DIV 0). In recent study, the concentrations lie within the 0.4–64 nmol TCS/mg tissue protein levels may result from human dermal exposure to TCS-containing products. In addition, 0.8–2.8 nmol TCS/mg protein doses within cells/tissues induce adverse effects [52]. In mice, TCS was detected in brain the 12 h after application to the skin [53]. Thus, we hypothesis that TCS may accumulate in the brain at concentration 10^−8^ M. On DIV 1, cells were treated with vehicle controls (DMSO; 0.001%) and TCS at a low concentration 10^−8^ M or 10^−6^ M. On DIV 4, neurons were harvested for quantification of axon and dendrite morphology. For proliferative experiments, neurons were incubated with 5-bromo-3-deoxyuridine (BrdU, 10 mM/mL, Sigma-Aldrich, St. Louis, MO, USA) after 12 h treated with TCS. Then, neurons were harvest after 2–3h BrdU-treatment.

#### Neuronal Morphology

Axons were identified with the axonal marker Tau1 and dendrites were identified with the dendritic marker MAP2. For axons: the longest axon was recognized as the primary axon; the remainder was considered axonal branches. The secondary axons grow branching point in the primary axon. The length of primary axon was determined starting from the base of an axon (the site attached to the soma) to the tip. For dendrites: the primary dendrites are the processes directly emerging from the soma. The secondary dendrites grow branching point in the primary dendrite. The total dendrite length, including primary dendrites and all dendritic branches. Measurements of axon and dendrite length were carried out using the ImageJ software (NIH, Bethesda, MD, USA).

### 4.3. Immunofluorescence

#### 4.3.1. Staining without BrdU

Neuronal cells were fixed in 4% formaldehyde (PFA) and then permeabilized with phosphate buffered saline (PBS) containing Triton X-100 (Sigma-Aldrich) (0.01% for cell). Then, neuron cells were blocked in PBS ++ (PBS + 5% goat serum (Vector Laboratories, Burlingame, CA, USA) + 0.25% Triton X-100) for 1h, followed by incubation in primary antibodies (Cleaved caspase-3, Cell Signaling Technology, Danvers, MA, USA, cat. no. 9661S, 1:400; MAP2, Abcam, Cambridge, UK, cat. no. ab32454, 1:500; and Tau1, Abcam, cat. no. ab75714, 1:500) at 4 °C overnight. For secondary staining, cells were incubated for 1h in secondary antibody solution (Alexa Fluor488 goat anti-rabbit IgG, cat. no. A11034, 1:1000, Alexa Fluor594 goat anti-rabbit, cat. no. A11012, 1:1000 and Alexa Fluor488 goat anti-chicken, cat. no. A11039, 1:1000, Invitrogen, Carlsbad, CA, USA) that contain 100 ng/mL 4′,6-diamidino-2-phenylindole (DAPI) (Sigma-Aldrich). Then cells were mounted in Flouro-Gel (Emsdiasum, Hatfield, PA, USA).

#### 4.3.2. BrdU Staining

BrdU staining was performed as described previously [54]. Neuronal cells were fixed in 4% PFA for 10 min at room temperature (RT). After permeabilized with PBS containing Triton X-100, neuron cells were incubated in 1 M HCl (Daejung Chemicals & Metals Co., Ltd., Gyeonggi-do, Korea) for 30 min at RT. Then, cells were neutralized in 0.1 M borate buffer (Sigma-Aldrich) for 30 min. Then, cells were blocked PBS++ for 1h. For BrdU immunolabeling, cells were then incubated in the appropriate primary antibodies BrdU (BD Bioscience, Durham, NC, USA, cat. no. 555627, 1:1000) at 4 °C overnight. Then, cells were incubated for 1h in secondary antibody solution (Alexa Fluor594 goat anti-rabbit contain DAPI). Following secondary antibody incubation and then were mounted in Fluoro-Gel. Fluorescently labelled cells were visualized using confocal microscopy.

### 4.4. RNA Extraction and Quantitative Real-Time PCR

Total RNA was extracted from whole brain, thyroid and pituitary gland of each mice by using Trizol reagents (Ambion, Austin, TX, USA) according to the manufacturer’s protocols. cDNA synthesis was performed as previously described [24]. Quantitative real-time PCR analysis was carried out in a QuantStudio 3 (Applied Biosystems, Foster City, CA, USA). GAPDH served as internal control. The primer sequences are presented in Appendix A.

### 4.5. Western-Blot Analysis

Brain total protein content was extracted using Pro-prep solution (iNtRON, Seoul, Korea) according to the manufacturer’s protocol. 100 g of protein was resolved by using 12% sodium dodecyl sulfate-polyacrylamide gel electrophoresis and transferred to polyvinylidene fluoride membrane (Merck Millipore, Taunton, MA, USA) as previously described [55]. Then, the membrane was incubated overnight in primary antibodies (Cleaved caspase-3, Cell Signaling Technology, cat. no. 9661S, 1:400; Bcl2, Santa Cruz Biotechnology, Santa Cruz, CA, USA, cat. no. 7382, 1:1000) and secondary antibodies (anti-rabbit, Cell Signaling Technology, cat. no. 7074S, 1:3000; anti-mouse, Cell Signaling Technology, cat. no. 7076P2, 1:3000). Membranes were enhanced using chemiluminescence reagent (EMD Millipore Corporation, Burlington, MA, USA). The optical density of the target band was detected with the Chemi Doc equipment, GenGnome5 (Syngene, Cambridge, UK) and analyzed by using Image J software.

### 4.6. Behavioral Analysis

#### 4.6.1. Experimental design. 

At 6 and 10 week-age, offspring mice were randomly selected to perform behavioral tests as described [51]. On testing days, mice were transferred to the testing room for at least 30 min before the start and the testing were performed by laboratory technicians blinded to the mouse group information. All experiments were performed between 8:00 AM and 2:00 PM and a resting period of 2 days to 1 week was used between two consecutive tests. All experimental areas were cleaned with 70% ethanol before the tests and between subjects.

#### 4.6.2. Open Field Test

The open filed test performed in a large acrylic cube measuring by 50 cm tall, 60 cm wide with a white bottom. Briefly, mice were individually placed near the wall-side and allowed to freely move for 5 min to measure locomotor activity. The movement of mice was recorded and analyzed with EthoVision XT14 software (Noldus, Leesburg, VA, USA). Time spent in the center zone (15 × 15 cm imaginary square), velocity and distance travelled was evaluated.

#### 4.6.3. Morris Water Maze

The Morris water maze test was conducted in a circular pool (90 cm diameter, 40 cm high), filled with water (25 ± 1 °C) and the water was made opaque by adding skim milk. The tank was divided into four equally sections (I, II, III, IV). A circular platform (10 cm diameter, 20 cm high) made of plexiglass was placed in the middle of a target quadrant (III) (12 cm from the pool’s edge and 1 cm below the surface of the water) with visual cues on the pool walls as spatial references. Mice received a two-phase training protocol for nine days: cue training (four days) and followed by spatial training (five days); four trials were performed per mouse per day and the escapes latency (time to find the hidden platform) in each trial was recorded. For each trial, the subject mice were gently placed into the water facing the wall from one of three quadrants (I, II and IV), varied by day of testing. Then, mice were given 60 s to find platform and a trial completed when the mice had found the platform. If the mouse failed to find the platform within 60 s period, it was gently guided onto the platform by the experimenter and allowed to rest on it for 30 s. The mean of escape latencies (second) for 4 trials is represented for the learning result for each mouse. On day 10 (probe test day), the platform was removed from the pool, and the mice were left to search for the platform for 60 s. Videos were recorded and analyzed using EthoVision XT14, the time spent in the target platform and the number of crossing the platform were measured to indicate the memory results.

#### 4.6.4. Novel Object Recognition

The first, the subject mice were placed into the open field arena in the presence of two identical objects (2 cm width × 5 cm length × 9 cm tall) and freely explored for 10 min. After 6 h, one of the objects was replaced by a novel object (different shape and color compared to old object) and the subject mice were placed inside the field to explore for another 10 min. The duration of time mice spent to interact with the novel and old objects (sniffing or exploring at distance within 2 cm of the object) were recorded. Videos were analyzed using EthoVision XT14.

#### 4.6.5. Tail Suspension Test

Each mouse was suspended on the edge of a shelf, 50 cm above the surface of a table. The subject mice were allowed to move for 6 min, and behavior was recorded by a camera. Videos were analyzed using EthoVision XT14. The duration of immobility was recorded in the last 5 min.

#### 4.6.6. Forced Swimming Test

Each mouse was gently placed into a glass cylinder (20 cm height,15 cm in diameter) filled with water (25 ± 2 °C) to a depth of 12 cm. All mice were forced to swim for 5 min and the duration of immobility was recorded. Videos were analyzed using EthoVision XT14.

#### 4.6.7. Three-Chamber Social Test

The three-chambered social test was performed as follows [56]. The three-chambered apparatus consisted of three Plexiglas chambers, each chamber was 20 cm width × 40 cm length × 22 cm tall and the dividing walls have small square openings (10 × 5 cm) allowing mice free access into each chamber. Both side chambers contained a cylindrical plastic cage (17 cm in height, a bottom diameter of 8 cm with the bars spaced 1 cm apart) in the corner that used to hold the stranger mice. The first, the subject mice were allowed to freely explore all three chambers with an empty plastic cage in each side chamber for the 5 min habituation period. For sociability testing, an unfamiliar C57BL/6 J mouse (Stranger 1- ‘S1′) was then introduced in a cylindrical plastic cage in one of the side-chambers and an empty cylindrical plastic cage (Empty- ‘E’) on the other side-chamber. Then, the subject mouse was placed in the center chamber and allowed to freely explore all three chambers for 10 min. For social novelty test, the empty plastic cage was replaced with a wild-type stimulus mouse (Stranger 2- ‘S2′) and the subject mouse again freely explored all three chambers for 10 min. All stranger mice were same sex at the same age with the subject mice and previously habituated to the plastic cages. Time spent in close proximity, distance travelled and heat maps were calculated using the automated software EthoVision XT14. The preference index for each animal was calculated as: Preference index = $\frac{\left(S1-E\right)}{\left(S1+E\right)}$ or as $\frac{\left(S2-S1\right)}{(S2+S1}$: where ‘E’, ‘S1′ and ‘S2′ are the time spent in close proximity with empty cage, the stranger animal 1 and 2, respectively.

#### 4.6.8. Nest-Building Test

The nest-building behavior of the mice was assayed by assessing the nest quality after 12 h exposure to a nestlet of tissue cotton (5 × 5 cm, mean weight 2.5 g). Each mouse was placed in its one clean cage with a nestlet at 19:00. Nest-building ability was assessed according to a 5-point rating scale from 1 to 5 as follows [57]. All mice were assessed together and the mean score of each group were determined.

### 4.7. Statistical Analysis

All statistical analyses were performed by applying two-way ANOVA, unpaired Student’s t tests for two population comparisons or one-way ANOVA (Bonferroni’s multiple comparison test). Data were randomly collected and analyzed using GraphPad Prism software (GraphPad Software, La Jolla, CA, USA). The results are presented as means ± SEM and the *p* values for each comparison are described in the results section. Each experiment in this study was performed blind and randomized. Animals were assigned randomly to the various experimental groups, and data were collected and processed randomly. The allocation, treatment, and handling of animals were the same across study groups. The number of mice and statistical detail for all behavioral assays are described in Appendix A. All treatment group results were compared to the vehicle group and each other.

## Figures and Tables

**Figure 1 ijms-21-04009-f001:**
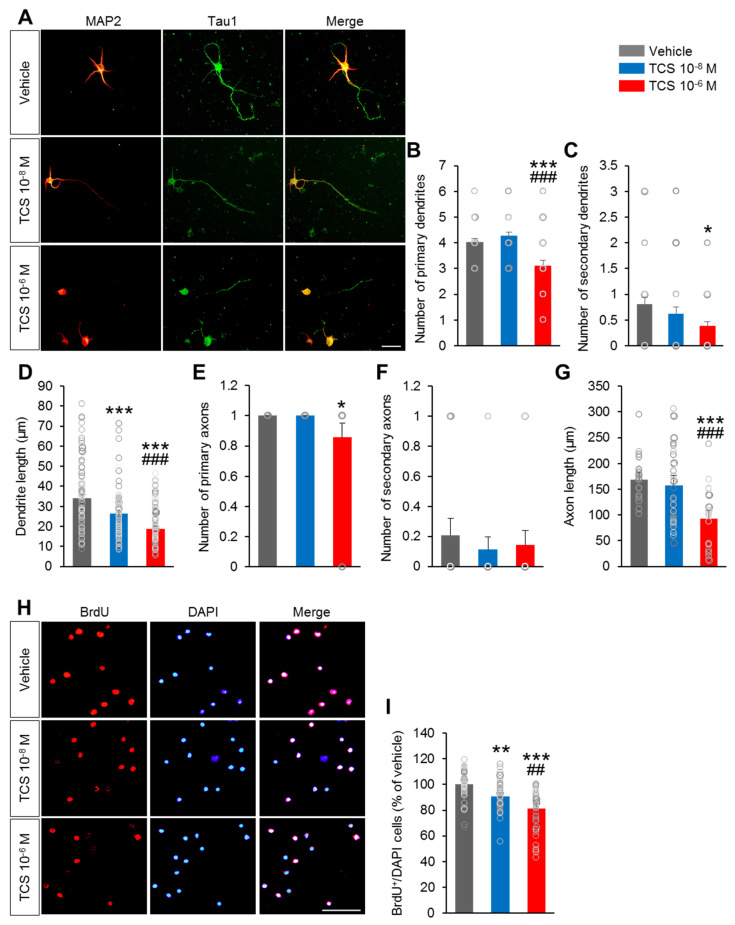
TCS impair to axon and dendrite growth. (**A**) Representative images of cultured mouse cortical neuron fluorescently labeled for MAP2 and Tau1. Scale bar, 100 µm. (**B**–**D**) Quantification of primary and secondary dendrite. (**B**) Number of primary dendrites, (**C**) number of secondary dendrites, (**D**) Average dendrite length. TCS 10^−8^ M group showed a markedly decreased the length of primary dendrites but did not change the number of primary dendrites and secondary dendrites compared to vehicle group. TCS 10^−6^ M group exhibited the significantly lower in the number of primary dendrites. The length of primary dendrites was also decreased in the TCS 10^−6^ M group compared to the vehicle group. (**E**–**G**) Quantification of primary and secondary axon. (**E**) Number of primary axons, (**F**) number of secondary axons, (**G**) average axon length. TCS 10^−8^ M group showed no significant difference in the number of primary and secondary axons. The length of axon and the number of primary axons were also decreased in TCS 10^−6^ M compared to the vehicle group. *n*  =  5 cell culture replicates using 5 mice for each condition (cell counts: 250 cells for each group). (**H**) Primary cortical neuronal cells were cultured from E15 mice and treated TCS at DIV 1. The cells were then incubated with 10 μM BrdU for 2-4 h after 12 h adding TCS, fixed and immunofluorescence with BrdU antibodies. Scale bar, 40 μm. (**I**) Quantification of A. TCS-treated groups exhibited lower in the percentage of cleaved BrdU^+^ cells compared to vehicle group. The percentage of cleaved BrdU^+^ cells in the TCS 10^−6^ M group was significantly lower compared to the TCS 10^−8^ M group. (J) Cell death was assessed in vehicle and TCS-treated groups by immunofluorescence using a cleaved caspase-3 antibody on DIV 2. Scale bars, 100 μm. (**K**) Quantification of **C**. The percentage of cleaved caspase-3^+^ cells was markedly higher in TCS 10^−8^ M and TCS 10^−6^ M groups. The percentage of cleaved caspase-3^+^ cells in the TCS 10^−6^ M group was higher compared to the TCS 10^−8^ M group (cell counts: 1,000 cells for each group). Data represent mean ± SEM. Statistical significance was determined by one-way ANOVA with Bonferroni correction. * *p* < 0.05 vehicle vs. TCS, ** *p* < 0.01 vehicle vs. TCS, *** *p* < 0.001 vehicle vs. TCS, ^##^
*p* < 0.01 TCS 10^−8^ M vs. TCS 10^−6^ M, ^###^
*p* < 0.001 TCS 10^−8^ M vs. TCS 10^−6^ M. Treatments: vehicle; 0.1% DMSO, TCS; 10^−6^ mol/l or 10^−8^ mol/L.

**Figure 2 ijms-21-04009-f002:**
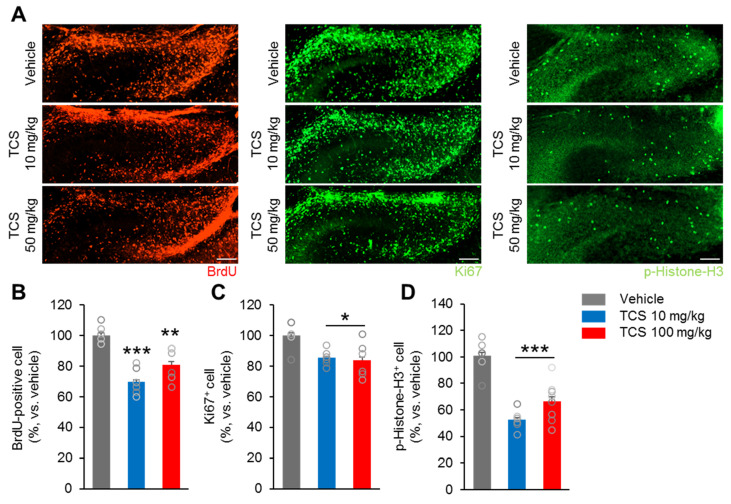
TCS-exposure alter neurogenesis in in vivo. (**A**) Cortical sections from from vehicle and TCS-treated groups at E18.5 were immunostained using BrdU antibody (red), Ki67 (green) and phospho-histone H3 antibody (green). (**B**–**D**) Quantification analysis of **A**. The number of BrdU^+^, Ki67^+^ and p-Histone-H3^+^ cells were decreased in the TCS-treated groups compared to vehicle group. However, there was no difference in the number of BrdU^+^, Ki67^+^ and p-Histone-H3^+^ cells between TCS 10 mg/kg and 100 mg/kg groups (*n* = 5 per group). (**E**) The cell cycle re-entering was calculated by dividing the number of BrdU-positive/Ki67-positive cells by the total number of BrdU-positive cells. Arrows indicate re-entry cell cycle cells (BrdU-positive/Ki67-positive). (**F**) TCS-treated groups exhibited lower number of cells re-entering the cell cycle compared to the vehicle group. Data represent mean ± SEM. Statistical significance was determined by one-way ANOVA with Bonferroni correction. * *p* < 0.05 vehicle vs. TCS, ** *p* < 0.01 vehicle vs. TCS, *** *p* < 0.001 vehicle vs. TCS. Treatments: vehicle; 0.1% DMSO, TCS; 10^−6^ mol/L or 10^−8^ mol/L.

**Figure 3 ijms-21-04009-f003:**
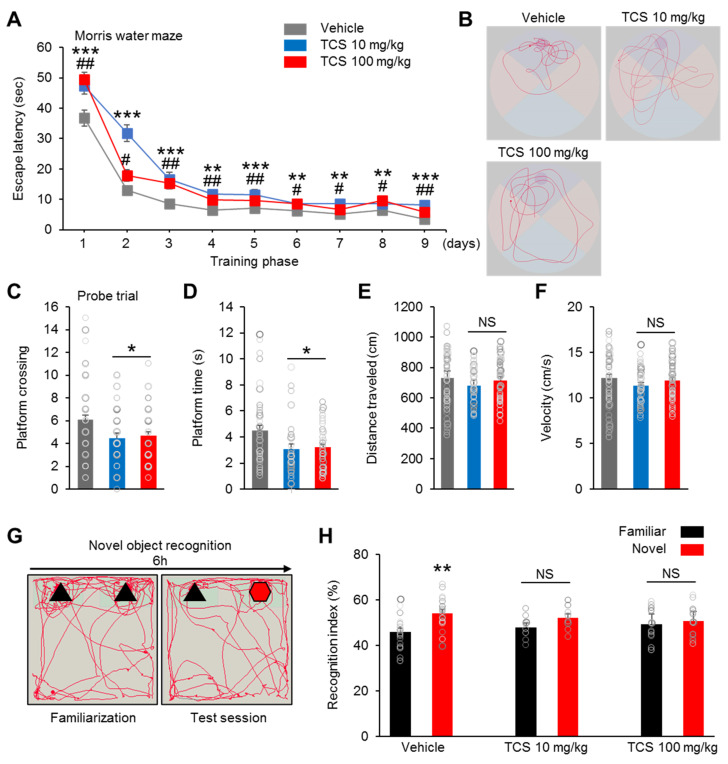
TCS impaired learning and recognition memory in mice. (**A**) In Morris water maze test, TCS-treated groups exhibit spatial learning disability; *n* = 18 mice (7 male, 11 female) for vehicle, 11 mice (8 male, 3 female) for TCS 10 mg/kg, 17 mice (10 male, 7 female) for TCS 100 mg/kg. (**B**) Representative swimming paths of vehicle, TCS 10 mg/kg and TCS 100 mg/kg during a probe trial after training. (**C**–**F**) Quantification of **B**. TCS 10 mg/kg and TCS 100 mg/kg group showed a decrease in platform crossings and time spent in platform. TCS-treated groups showed no change in swimming distances and swimming speeds. * *p* < 0.05 vehicle vs. TCS 10 mg/kg, ** *p* < 0.01 vehicle vs. TCS 10 mg/kg, *** *p* < 0.001 vehicle vs. TCS 10 mg/kg. ^#^
*p* < 0.05 vehicle vs. TCS 100 mg/kg, ^##^
*p* < 0.01 vehicle vs. TCS 100 mg/kg. NS, no significance. (**G**) Schematic diagram of the novel object recognition test. (**H**) Novel object test shows recognitive impairment in TCS-treated groups. *n* = 18 mice (7 male, 11 female) for vehicle, 9 mice (6 male, 3 female) for TCS 10 mg/kg, 14 mice (8 male, 6 female) for TCS 100 mg/kg. Data represent mean ± SEM. Statistical significance was determined by two-way ANOVA. ** *p* < 0.01 familiar vs. novel object. NS, no significance. Treatments: corn oil; vehicle, TCS; 10 mg/kg/day, TCS; 100 mg/kg.

**Figure 4 ijms-21-04009-f004:**
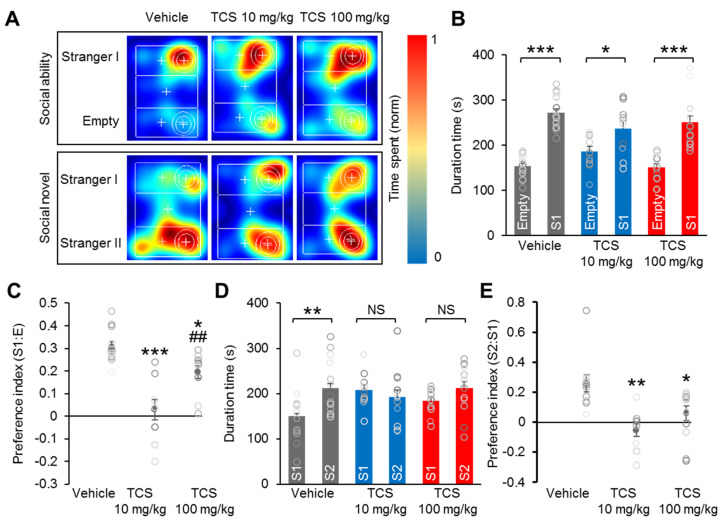
TCS mice display social behavioral alteration in three-chamber tests. (**A**) Representative heat map images from ‘Stranger I–Empty’ and ‘Stranger I–Stranger II’ from social ability and novel tests. (**B**,**C**) Quantification of **A** for sociability test. TCS-treated groups interact with ‘Stranger’ stimulus mice. Both vehicle and TCS-treated groups display preference for the chamber containing an unfamiliar mouse ‘S1′ than in the empty chamber, but the TCS-treated group preference index was significantly reduced compared to vehicle group. (**D**,**E**) Quantification of **A** for social novelty test. TCS-treated groups showed no preference for a stranger mouse ‘S2′ over a familiar mouse ‘S1′ and lower index of preference for the novel stimulus than in vehicle group. Data represent mean ± SEM. Statistical significance was determined between times exploring empty chamber and stranger I, or times exploring stranger I and stranger II for each group and condition by a two-tailed Student’s t test. For preference index, statistical significance was determined by one-way ANOVA with Bonferroni correction. * *p* < 0.05 and ** *p* < 0.01 vs. vehicle, *** *p* < 0.001 vs. vehicle, ^##^
*p* < 0.01 TCS 10 mg/kg vs. TCS 100 mg/kg. NS, no significance. *n* = 18 mice (7 male, 11 female) for vehicle, 11 mice (8 male, 3 female) for TCS 10 mg/kg, 17 mice (10 male, 7 female) for TCS 100 mg/kg. Treatments: corn oil; vehicle, TCS; 10 mg/kg/day, TCS; 100 mg/kg.

**Figure 5 ijms-21-04009-f005:**
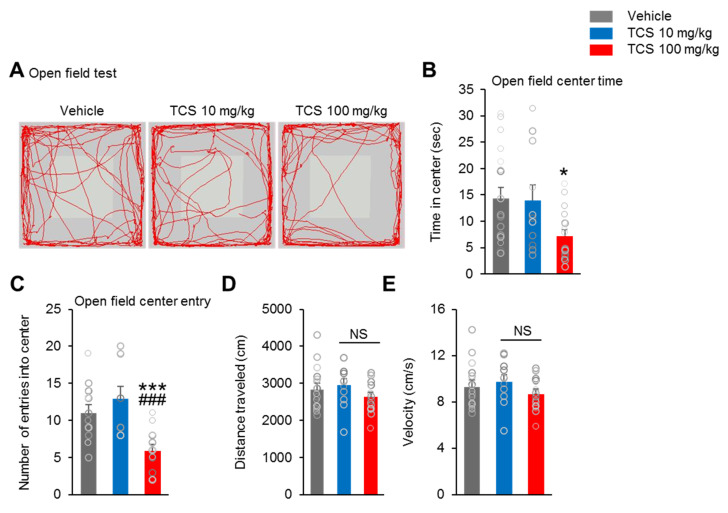
TCS impair locomotor activity, anxiety-like behavior and nesting- behavior in mice. (**A**) Representative tracing of mouse travel in the open field test. (**B**–**E**) Quantification of **A**. The time spent in the center, the number of entries into the center, distance traveled and velocity in the open field were examined in the open field. TCS 100 mg/kg group showed reduction of both open field center time and open field center entry compared to vehicle and TCS 10 mg/kg group. TCS-treated groups exhibited no difference in distance traveled and velocity in open field. *n* = 18 mice (seven male, 11 female) for vehicle, 11 mice (eight male, three female) for TCS 10 mg/kg, 17 mice (10 male, seven female) for TCS 100 mg/kg. * *p* < 0.05 vs. vehicle, *** *p* < 0.001 vs. vehicle, ^###^*p* < 0.001 TCS 10 mg/kg vs. TCS 100 mg/kg. NS, no significance. (**F**) In tail suspension test, TCS-treated group showed no difference in the immobility times compared to vehicle group. (**G**) In forced swimming test, there was no changes in the immobility times between vehicle and TCS-treated groups. *n* = 18 mice (seven male, 11 female) for vehicle, 11 mice (eight male, three female) for TCS 10 mg/kg, 17 mice (10 male, seven female) for TCS 100 mg/kg. Data represent mean ± SEM. Statistical significance was determined by one-way ANOVA with Bonferroni correction. NS: no significance.

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
