# Peer review of "Perinatal Exposure to Triclosan Results in Abnormal Brain Development and Behavior in Mice"

_ijms, 2020, doi:10.3390/ijms21114009_

Round 1

Reviewer 1 Report

The authors present in a general clear way the detrimental effect of perinatal exposure to TCS in mice.

Over all, the experiments are clear and well performed showing a real impact in the litters from TCS-exposed pregnant mice. There are some weak point to be considered as better explain the treatment details, which are not very clear. Moreover, several minor mistakes should be corrected.

After clarifying these points and solving some minor the minor mistakes I would consider this paper for publication.

Author Response

We would like to thank the reviewers for their constructive suggestions on how to improve the quality of this manuscript. In the revised version, we have addressed the concerns that the reviewers raised.

Reviewer #1

The authors present in a general clear way the detrimental effect of perinatal exposure to TCS in mice.

Overall, the experiments are clear and well performed showing a real impact in the litters from TCS-exposed pregnant mice. There are some weak point to be considered as better explain the treatment details, which are not very clear. Moreover, several minor mistakes should be corrected.

After clarifying these points and solving some minor the minor mistakes I would consider this paper for publication.

Response: We would like to thank you for your comment. In the present study, the maternal mice were randomly divided into three groups including vehicle group and two exposure groups (n = 5 mouse/group, each mouse was maintained in each cage) and they chronically received different TCS levels by subcutaneous injection. Mice in the three groups were given 0.2 mL of corn oil with TCS concentrations of 0, 10 and 100 mg/kg every day from E9.5 to postnatal day (PND) 28.5. We have added this information in line 409 to 413.

Reviewer 2 Report

The study by Tran et al. examined effects of triclosan (TCS) on the brain development. Evidence shows that TCS may impair brain development, thus leading to behavioural effects.

General comments:

  1. In Wang et al. paper in Scientific Reports TCS had severe consequences already at a dose of 10 mg/kg. What was the abortion rate in the current study?
  2. TCS was used in concentrations of 10 nM and 1 μM. What concentrations could be expected under in vivo conditions?

Introduction:

  1. “TCS has also been associated with low birth weight suggest that TCS is able to cross the placenta barrier, leading the fetal TCS exposure” Does low birth weight really indicate that TCS crosses the placenta? An indirect is also possible.
  2. Abbreviation for triclosan (TCS) is used inconsistently. Sometimes the full name is used and sometimes TCS within the same paragraph.
  3. Changes in fetal hypothalamic transcriptomics are mentioned. Probably changes in transcriptome not transcriptomics?

Results and methods:

  1. “However, the number of secondary dendrites (F2,147=3.594, p=0.0300) compared to the vehicle group (Figure 1B and 1C).” Something missing here? “We found that dendrite length and numbers of primary and secondary dendrites were determined at DIV4 (Figure 1A).” It is not clear what is meant here. There are several other statements like this in the manuscript. The paper would gain if it is revised by a native speaker.
  1. Many bands were obtained with the caspase 3 antibody. Was this really a cleaved caspase 3 antibody or does it also detect the full length caspase 3? This is also important for interpretation of the immunocytochemistry data.
  2. Calculations for assessment of proliferation should be described in the methods. What does 100% mean in Figure 1-1 I for instance. (Were all control cells BrdU+?)
  3. Assessment of apoptosis with immunocytochemistry should be described in the methods. Were there any positive controls for apoptosis? What does 100% mean in Figure 1K?
  4. Was necrosis also assessed?
  5. “These results suggest that TCS can inhibit proliferation and promote apoptosis of neuronal progenitor cells during brain development.” Apoptosis was not assessed in vivo.
  6. What does OP-exposure mean in the title of Figure 2? OP is usually an abbreviation for organophosphates.
  7. If phospho-Histone H3 was assessed the figure should indicated so (the y-axis title uses the term Histone-H3, which would mean total or dephosphorylated histone H3.)
  8. Dental gyrus or dentate gyrus?
  9. Dendrite and axon quantification should be described in the methods. Primary/secondary axons/dendrites should also be defined.
  10. Methods should also report catalogue numbers of the antibodies used in the study.
  11. Gene expression was used to estimate whether TCS interfered with the action of thyroid hormone. Was TSH, T4 or T3 altered in plasma? This would provide a more direct evidence of interference with the thyroid function.
  12. TSH was apparently not assessed on the transcriptional level, although this would provide further evidence that secretion of TRH and TSH might have been altered.
  13. Antibody against Bax is mentioned in the methods but Bax does not appear anywhere in the text.

Author Response

We would like to thank the reviewers for their constructive suggestions on how to improve the quality of this manuscript. In the revised version, we have addressed the concerns that the reviewers raised.

The study by Tran et al. examined effects of triclosan (TCS) on the brain development. Evidence shows that TCS may impair brain development, thus leading to behavioural effects.

General comments:

  1. In Wang et al. paper in Scientific Reports TCS had severe consequences already at a dose of 10 mg/kg. What was the abortion rate in the current study?

Response: First, we would like to thank you for your comment. In the current study, we did not determine the effects of TCS on the abortion rate. However, we found that perinatal exposure to TCS significantly decreased the number of pups. We added more information in the result in lines 165-167 and discussion at line 345.

  1. TCS was used in concentrations of 10 nM and 1 μM. What concentrations could be expected under in vivoconditions?

Response: Recently study, the concentrations lie within the 0.4-64 nmol TCS/mg tissue protein levels may result from human dermal exposure to TCS-containing products. In addition, 0.8–2.8 nmol TCS/mg protein doses within cells/tissues induce adverse effects (Weatherly and Gosse, 2017). In mice, TCS was detected in brain the 12 hours after application to the skin (Fang et al., 2016). In in vitro experiment, primary neuronal cells were exposed to TCS at a range dose 10-5 M, 10-6 M, 10-7 M, 10-8 M, and 10-9 M. However, at high dose (10-5 M) we found that all most of neuron cells were death (data not shown). And at a low dose (10-9 M), there was no change in the morphology of neuron cells compared to vehicle (data not shown). Thus, primary neuronal cells were received TCS at concentrations 10-8 M or 10-6 M in this study. We have added this information in line 427 to 432.

Introduction:

  1. “TCS has also been associated with low birth weight suggest that TCS is able to cross the placenta barrier, leading the fetal TCS exposure”. Does low birth weight really indicate that TCS crosses the placenta? An indirect is also possible.

Response: As the reviewer indications, low birth weight may not really indicate that TCS crosses the placenta. We have changed this sentence by “Moreover, TCS can cross the placental barrier, leading the fetal TCS exposure.” in line 54-55.

  1. Abbreviation for triclosan (TCS) is used inconsistently. Sometimes the full name is used and sometimes TCS within the same paragraph.

Response: We have changed “triclosan” to “TCS” in all relevant sections of the text.

  1. Changes in fetal hypothalamic transcriptomics are mentioned. Probably changes in transcriptome, not transcriptomics?

Response: Following the reviewer’s advice, we have changed “transcriptomics” to “transcriptome” at line 72 and 391.

Results and methods:

  1. “However, the number of secondary dendrites (F2,147=3.594, p=0.0300) compared to the vehicle group (Figure 1B and 1C).” Something missing here? “We found that dendrite length and numbers of primary and secondary dendrites were determined at DIV4 (Figure 1A).” It is not clear what is meant here. There are several other statements like this in the manuscript. The paper would gain if it is revised by a native speaker.

Response: Thank you for your comment. We apologize for these errors. We have fixed these errors by changing:

In line 101 to 103: “We assessed the dendrite length and numbers of primary and secondary dendrites at DIV4 (Figure 1A). At the TCS low dose (10-8 M), TCS showed no effect on the number of primary dendrites compared to that in the vehicle group (F2,150=16.69, p<0.0001) (Figure 1B).”

In line 103 to 106: “However, the number of secondary dendrites (F2,147=3.594, p=0.0300) in TCS 10-6 M group were significantly decreased compared to the vehicle group (Figure 1C). The length of dendrites was significantly decreased in the TCS 10-8 M group compared with the vehicle group (F2,440=43.93, p<0.0001) (Figure 1D).”

In line 118 to 120: “In addition, the number of primary axons was also smaller in the TCS 10-6 M group (F2,150=7.594, p=0.0007) compared to that in the vehicle group (Figure 1F)”.

  1. Many bands were obtained with the caspase 3 antibody. Was this really a cleaved caspase 3 antibody or does it also detect the full length caspase 3? This is also important for interpretation of the immunocytochemistry data.

Response: Following the antibody information (Cat no. 9661) of Cell Signaling Technology, we detected the cleaved 17/19 kDa fragment of activated caspase-3 by Western blot and immunocytochemistry. This antibody has been generally used by detecting the activated caspase-3 (Cho et al., 2018; Santana-Codina et al., 2018).

  1. Calculations for assessment of proliferation should be described in the methods. What does 100% mean in Figure 1-1 I for instance. (Were all control cells BrdU+?)

Response: As the reviewer suggested, the calculations for assessment of proliferation are described: “The percentage of BrdU+/DAPI cell was quantified”. However, we have added this information in the Results part in line 129 to avoid complex sentences in the Method part. And the 100% means: “The percentage of BrdU+/DAPI cell in TCS-treated groups compared to vehicle group”. We also changed “BrdU+ cells (%, per DAPI)” to “BrdU+/DAPI cells (% of the vehicle)” in figure 1I.

Also, we also added the description of the calculations for assessment of the number of BrdU+, Ki67+ and p-Histone-H3+ cells in the dentate gyrus at line 145. “The number of BrdU+, Ki67+ and p-Histone-H3+ cells in the DG were determined.

  1. Assessment of apoptosis with immunocytochemistry should be described in the methods. Were there any positive controls for apoptosis? What does 100% mean in Figure 1K?

Response: Again, we have added the description of the calculations for assessment of apoptosis: “The percentage of cleaved caspase-3+/DAPI cell was quantified”. However, we added this information in the Results part in line 135-136 to avoid complex sentences in the Method part. And the 100% means: “The percentage of cleaved caspase-3+/DAPI cell in TCS-treated groups compared to vehicle group”. We also so change the “cleaved caspase-3+ cells (%, per DAPI)” to “cleaved caspase-3+/DAPI cells (% of the vehicle)” in figure 1K.

  1. Was necrosis also assessed?

Response: Sorry, we did not assess necrosis due to there was no change in the morphology of the brain and the inflammation in the brain. For example, histopathologic changes found in radiation necrosis (include coagulative and liquefactive necrosis predominately in the white matter) or the infiltration of inflammatory cells in the brain (Yoshii, 2008).

  1. “These results suggest that TCS can inhibit proliferation and promote apoptosis of neuronal progenitor cells during brain development.” Apoptosis was not assessed in vivo.

Response: We appreciate the reviewer for clarifying this point. We have assessed apoptosis in the brain of adult offspring mice at PND119. We found that TCS-treated groups showed significantly higher cleaved caspase-3 levels compared with the vehicle group. Additionally, the protein levels of anti-apoptotic Bcl2 were significantly lower in the TCS-treated groups compared to the vehicle group. This result showed supplementary figure 1F and 1G.

  1. What does OP-exposure mean in the title of Figure 2? OP is usually an abbreviation for organophosphates.

Response: Again, we apologize for this error. We have fixed this error by changing “OP” to “TCS” at line 754.

  1. If phospho-Histone H3 was assessed the figure should indicated so (the y-axis title uses the term Histone-H3, which would mean total or dephosphorylated histone H3.)

Response: As the reviewer’s indication, we have changed “Histone-H3” to “p-Histone-H3” in figure 2D.

  1. Dental gyrus or dentate gyrus?

Response: We have changed “dental” to “dentate” at line 143.

  1. Dendrite and axon quantification should be described in the methods. Primary/secondary axons/dendrites should also be defined.

Response: As the reviewer suggested, we added the dendrite and axon quantification in the Methods part from line 436 to 444.

Line 436: “Neuronal morphology

Axons were identified with the axonal marker Tau1 and dendrites were identified with the dendritic marker MAP2. For axons: the longest axon was recognized as the primary axon; the remainder was considered axonal branches. The secondary axons grow branching points in the primary axon. The length of the primary axon was determined to start from the base of an axon (the site attached to the soma) to the tip. For dendrites: the primary dendrites are the processes directly emerging from the soma. The secondary dendrites grow branching points in the primary dendrite. The total dendritic length, including primary dendrites and all dendritic branches. Measurements of axon and dendrite length were carried out using ImageJ software (NIH, Bethesda, MD, USA).”

  1. Methods should also report catalogue numbers of the antibodies used in the study.

Response: As the reviewer suggested, we added the catalogue numbers of the antibodies in the Methods part.

  1. Gene expression was used to estimate whether TCS interfered with the action of thyroid hormone. Was TSH, T4 or T3 altered in plasma? This would provide a more direct evidence of interference with the thyroid function.

Response: We did not investigate the effects of TCS on the level of TSH, T4, or T3 in plasma. However, the decrease of T3, T4, and TSH were confirmed by the reduction of Pax8/Nkx2.1 expression associated with their reduced transcriptional activity might contribute to the decreased expression of genes/proteins involved in the TH synthesis and secretion (Tshr, Nis, Tpo, Tg and Mct8) (Serrano-Nascimento et al., 2017). In this study, we found that perinatal exposure to the TCS reduction the expression of Pax8/Nkx2-1 associated with the reduction of their transcriptional activity (Tshr, Tg, Tpo, Nis, and Mct8). These results suggest that the changes in the thyroid hormone-related genes may suggest that perinatal exposure to TCS resulted in the inactivation of thyroid synthesis, although the thyroid hormone levels were not measured as mentioned in line 380-384.

  1. TSH was apparently not assessed on the transcriptional level, although this would provide further evidence that secretion of TRH and TSH might have been altered.

Response: We assessed the transcriptional level of TSHR in Supplementary figure 2. The main function of thyrotropin (TSH) is to stimulate the synthesis and release of thyroid hormones, triiodothyronine (T3) and thyroxine (T4). In this study, the mRNA levels of thyrotropin receptor (Tshr) in the thyroid gland were significantly decreased by prenatal exposure to TCS as shown in line 187 to 190. We also added more information in line 380-382: “Furthermore, the hypothyroid condition was confirmed by the reduction of Pax8/Nkx2.1 expression associated with their reduced transcriptional activity might contribute to the decreased expression of genes/proteins involved in the TH synthesis and secretion (Tshr, Nis, Tpo, Tg and Mct8) [47].”

  1. Antibody against Bax is mentioned in the methods but Bax does not appear anywhere in the text.

Response: We apologize for these errors. We have deleted the antibody Bax information in this part.

Reference

Cho, Y.-E., Seo, W., Kim, D.-K., Moon, P.-G., Kim, S.-H., Lee, B.-H., Song, B.-J., Baek, M.-C., 2018. Exogenous exosomes from mice with acetaminophen-induced liver injury promote toxicity in the recipient hepatocytes and mice. Scientific reports 8, 1-13.

Fang, J.L., Vanlandingham, M., da Costa, G.G., Beland, F.A., 2016. Absorption and metabolism of triclosan after application to the skin of B 6 C 3 F 1 mice. Environ. Toxicol. 31, 609-623.

Santana-Codina, N., Roeth, A.A., Zhang, Y., Yang, A., Mashadova, O., Asara, J.M., Wang, X., Bronson, R.T., Lyssiotis, C.A., Ying, H., 2018. Oncogenic KRAS supports pancreatic cancer through regulation of nucleotide synthesis. Nature communications 9, 1-13.

Serrano-Nascimento, C., Salgueiro, R.B., Pantaleão, T., da Costa, V.M.C., Nunes, M.T., 2017. Maternal exposure to iodine excess throughout pregnancy and lactation induces hypothyroidism in adult male rat offspring. Sci. Rep. 7, 1-12.

Weatherly, L.M., Gosse, J.A., 2017. Triclosan exposure, transformation, and human health effects. Journal of Toxicology and Environmental Health, Part B 20, 447-469.

Yoshii, Y., 2008. Pathological review of late cerebral radionecrosis. Brain tumor pathology 25, 51.

Reviewer 3 Report

Tran et al., submitted a manuscript, in which they investigate the effects of perinatal exposure to Triclosan (TCS), evident in widely used products and suggested to represent an endocrine-discripting chemical (EDC), on neuronal development and behaviour.

Triclosan is suspected to elicit a number of harmful defects in human, including lowered birth weight and behavioural abnormalities, and acting neurotoxic.

Hence, detailed investigation of its effects is crucial, for which the manuscript deals with relevant questions.

The authors provide interesting, relevant and well-presented data, which however lack some cohesiveness. A more detailed phenotypic characterisation could help here, which would strengthen the manuscript and the conclusions.

Especially the link to the thyroid hormone data is both, leaky but very relevant.

Detailed advice is given below.

The authors present cell culture experiments of cortical neurons that were treated with TCS, as well as histological data of mice that were perinatally exposed to TCS.

The cell culture data show defects on morphology, altered proliferation and cell death rates, when treated withTCS. here two different concentrations were used, whereby it is not clear to me on what these concentrations rely and whether they would be equivalent to the contractions that are evident in the brain upon perinatal exposure.

Next, the authors analysed cell proliferation and cell death in mice at E18.5 from mothers that were subcutaneously injected with TCS (10 mg/kg and 100 mg/kg or vehicle). The 10 mg/kg refer to the TCS exposure „equivalent to those of spontaneous abortions“. Here it is unclear to me, in which tissue/samples of spontaneous abortions this was measured and whether similar values in the corresponding tissues of mice can be seen upon subkutan injection of the respective concentrations. The „physiological range“ of TCS applied to the animals is an issue, which needs to be clearly stated. Also the TCS concentrations in brains of E18.5 embryos should be determined, to know what is indeed getting into the brain, and whether these are concentrations corresponding to what was applied in cell culture.

Another issue I have with the brain analysis is that the authors here analysed the dentate gyrus. Why not the hippocampus and why not the cortex (after having already analysed cortical neurons in culture)? Especially as there seem to be effects on brain weight and the ratio of brain weight/ body weight (seen at later stages upon perinatal TCS), one would expect effects in many parts off the brain including the cortex.

Here, a more detailed analysis would be favourable, not only including the cortex, but to characterise the effect on proliferation more deeply. The authors see reduced numbers of Ki67 or BrdU positive cells. Why is that so? Do they die? or do they differentiate too much too early, shifting the balance from proliferative divisions to neurogenic divisions precociously? Here, a KI67/BrdU double labelling may help to check, how much cells have left the cell cycle at a certain stage.

One would need to determine either, whether the reduced brain weight at adult stages is a developmental effect of eg. precocious differentiation of progenitors at the expense of the progenitor pool or increased apoptosis. So, an intermediate stage (eg. early postnatal) could be useful.

Caspase stainings in adult cortical sections, to determine which brain region is affected, would also be helpful, to better understand the behavioural abnormalities seen in this comprehensive set of data provided in the last part of the manuscript.

Finally, the relation of the TCS treatment and the changes in transcription of thyroid-hormone -related genes expression is not clear to me, albeit it is interesting  and important to follow… Does this substance has gene regulatory potential? Or do the authors monitor transcriptional changes in response to the altered cellular effects? If Dio2 is increased, do we see more thyroid hormone in the brain, and so could the effects of TCS be rather indirect from changing the thyroid hormone levels in the brain than directly acting in the brain? Here, again, I have to stress that it is crucial to determine the TCS levels that indeed reach  the brain.

Taken together, I have three major points:

  1. Are the applied TCS concentrations „physiological“ and which concentration of TCS reaches the brain
  2. Are the effects due to TCS in the brain or to altered thyroid hormone levels?
  3. more detailed phenotyping: intermediate stage for brain weight, BrdU/Ki67 labelling in the cortex (at embryonic stage), caspase staining of adult brains

Minor issues:

the data in the paragraph:

„2.6. TCS-exposure induces anxiety-like behavior and depression behavior in mice“ 

do not show significant changes in regard to depression-like behaviour, so the second part of the headline is wrong

Author Response

We would like to thank the reviewers for their constructive suggestions on how to improve the quality of this manuscript. In the revised version, we have addressed the concerns that the reviewers raised.

  1. The cell culture data show defects on morphology, altered proliferation and cell death rates, when treated with TCS. here two different concentrations were used, whereby it is not clear to me on what these concentrations rely and whether they would be equivalent to the contractions that are evident in the brain upon perinatal exposure.

Next, the authors analysed cell proliferation and cell death in mice at E18.5 from mothers that were subcutaneously injected with TCS (10 mg/kg and 100 mg/kg or vehicle). The 10 mg/kg refer to the TCS exposure „equivalent to those of spontaneous abortions“. Here it is unclear to me, in which tissue/samples of spontaneous abortions this was measured and whether similar values in the corresponding tissues of mice can be seen upon subkutan injection of the respective concentrations. The „physiological range“ of TCS applied to the animals is an issue, which needs to be clearly stated. Also the TCS concentrations in brains of E18.5 embryos should be determined, to know what is indeed getting into the brain, and whether these are concentrations corresponding to what was applied in cell culture.

Response: First, we appreciate the reviewer for clarifying this point. As reviewer indications, we have added some changes at line 413-414: “In mice, the dose of 10 mg/kg TCS exposure caused an equivalent urinary TCS level as those in the high-TCS abortion patients [50].”

We have added more information in line 427 to 432. In recent study, the concentrations lie within the 0.4–64 nmol TCS/mg tissue protein levels may result from human dermal exposure to TCS-containing products. In addition, 0.8–2.8 nmol TCS/mg protein doses within cells/tissues induce adverse effects (Weatherly and Gosse, 2017). In addition, TCS was detected in brain the 12 hours after application to the skin of mice (Fang et al., 2016). In pre-experiment, primary neuronal cells were exposed to TCS at a range dose 10-5 M, 10-6 M, 10-7 M, 10-8 M, and 10-9 M. However, at high dose (10-5 M) we found that all most of neuron cells were death (data not shown). And at a low dose (10-9 M), there was no change in the morphology of neuron cells compared to vehicle (data not shown). Thus, primary neuronal cells were received TCS at concentrations 10-8 M or 10-6 M in this study.

  1. Another issue I have with the brain analysis is that the authors here analysed the dentate gyrus. Why not the hippocampus and why not the cortex (after having already analysed cortical neurons in culture)? Especially as there seem to be effects on brain weight and the ratio of brain weight/ body weight (seen at later stages upon perinatal TCS), one would expect effects in many parts off the brain including the cortex.

Response: We performed immunofluorescence using a mature neuronal marker (NeuN). However, we also did not find significant changes in neuronal density in the brain of TCS-treated groups compared to the vehicle group. Therefore, future work will focus on which neuronal cell type be affected in the brain. Please check the attached file included NeuN staining result on the last page. 

  1. Here, a more detailed analysis would be favourable, not only including the cortex, but to characterise the effect on proliferation more deeply. The authors see reduced numbers of Ki67 or BrdU positive cells. Why is that so? Do they die? or do they differentiate too much too early, shifting the balance from proliferative divisions to neurogenic divisions precociously? Here, a KI67/BrdU double labelling may help to check, how much cells have left the cell cycle at a certain stage.

Response: As the reviewer mentioned, we added the cell cycle re-entry results in the results part. We assessed the cell cycle regulation in the TCS-treated mice dentate gyrus. We found that there was a significantly lower proportion of cell cycle re-entry in the TCS-treated groups compared to the vehicle (BrdU+Ki67+/BrdU+) (Figure 2E and F) (F2,15=6.485, p=0.0093).  However, TCS 10 mg/kg group showed no significant difference in the proportion of cell cycle re-entry compared to the TCS 100 mg/kg group (Figure 2E and F). We have added more information in lines 153 to 158.

  1. One would need to determine either, whether the reduced brain weight at adult stages is a developmental effect of eg. precocious differentiation of progenitors at the expense of the progenitor pool or increased apoptosis. So, an intermediate stage (eg. early postnatal) could be useful.

Response: We apologize that all of the offspring mice had been sacrificed. Thus, the intermediate stage for brain weight is not available now. However, we found that brain weight at PND 119 significantly decreased in TCS 100 mg/kg group compared to the vehicle (Supplementary figure-1D). Thus, we predict that the brain weight decreases in TCS 100 mg/kg group in the intermediate stage.

  1. Caspase stainings in adult cortical sections, to determine which brain region is affected, would also be helpful, to better understand the behavioral abnormalities seen in this comprehensive set of data provided in the last part of the manuscript.

Response: We also performed the immunofluorescence using the caspase-3 antibody. However, no caspase-3 positive cell was observed in the adult offspring brain. Thus, we did western blot to assess caspase-3 protein content in the adult offspring brain. We found that the cleaved caspase-3 protein level increases in TCS-treated groups compared to vehicle groups. These results show in the supplementary figure 1E and 1F.

  1. Finally, the relation of the TCS treatment and the changes in transcription of thyroid-hormone -related genes expression is not clear to me, albeit it is interesting and important to follow… Does this substance has gene regulatory potential? Or do the authors monitor transcriptional changes in response to the altered cellular effects? If Dio2 is increased, do we see more thyroid hormone in the brain, and so could the effects of TCS be rather indirect from changing the thyroid hormone levels in the brain than directly acting in the brain? Here, again, I have to stress that it is crucial to determine the TCS levels that indeed reach the brain.

Response: The decrease of T3, T4, and TSH were confirmed by the reduction of Pax8/Nkx2.1 expression associated with their reduced transcriptional activity might contribute to the decreased expression of genes/proteins involved in the TH synthesis and secretion (Tshr, Nis, Tpo, Tg and Mct8) (Serrano-Nascimento et al., 2017). In this study, we found that perinatal exposure to the TCS reduction the expression of Pax8/Nkx2-1 associated with the reduction of their transcriptional activity (Tshr, Tg, Tpo, Nis, and Mct8). These results suggest that the changes in the thyroid hormone-related genes may suggest that perinatal exposure to TCS resulted in the inactivation of thyroid synthesis, although the thyroid hormone levels were not measured as mentioned in line 380 to 384.

Taken together, I have three major points:

  1. Are the applied TCS concentrations “physiological” and which concentration of TCS reaches the brain?

Response: Again, the concentrations lie within the 0.4–64 nmol TCS/mg tissue protein levels may result from human dermal exposure to TCS-containing products. In addition, 0.8–2.8 nmol TCS/mg protein doses within cells/tissues induce adverse effects (Weatherly and Gosse, 2017). We have added this information at line 427 to 432.

  1. Are the effects due to TCS in the brain or to altered thyroid hormone levels?

Response: We found that TCS can impair dendrite and axon growth by reducing average length and numbers of axons and dendrites using the in vitro model. Additionally, TCS inhibited the proliferation of and promoted apoptosis in neuronal progenitor cells by in vivo and in vitro models. We also found that TCS changed mRNA levels of thyroid hormone-related genes such as thyroglobulin (Tg), thyroperoxidase (Tpo), sodium-iodide symporter (Nis), monocarboxilate transporter 8 (Mct8) and thyrotropin receptor (Tshr) in the thyroid gland. Thus, we suggest that TCS may affected directly brain development and altered thyroid hormone functions. In the future, we will do a new experiment to assess the alteration of thyroid hormone function after TCS treatment during brain development.

  1. More detailed phenotyping: intermediate stage for brain weight, BrdU/Ki67 labelling in the cortex (at embryonic stage), caspase staining of adult brains.

Response: Thank you for your comment. To clarify these points, we added more information in lines 129, 135, 178, and 190.  We also added more information in supplementary figure at line 7, 8, and 10.

Minor issues:

the data in the paragraph:

“2.6. TCS-exposure induces anxiety-like behavior and depression behavior in mice” do not show significant changes in regard to depression-like behaviour, so the second part of the headline is wrong.

Response: Thank you for your comment. As the reviewer’s indication, we have fixed this error by changing “2.6. TCS-exposure induces anxiety-like behavior, but not affect depression-like behavior in mice” in line 260.

References

Fang, J.L., Vanlandingham, M., da Costa, G.G., Beland, F.A., 2016. Absorption and metabolism of triclosan after application to the skin of B 6 C 3 F 1 mice. Environ. Toxicol. 31, 609-62

3.

Serrano-Nascimento, C., Salgueiro, R.B., Pantaleão, T., da Costa, V.M.C., Nunes, M.T., 2017. Maternal exposure to iodine excess throughout pregnancy and lactation induces hypothyroidism in adult male rat offspring. Sci. Rep. 7, 1-12.

Weatherly, L.M., Gosse, J.A., 2017. Triclosan exposure, transformation, and human health effects. Journal of Toxicology and Environmental Health, Part B 20, 447-469.

Yoshii, Y., 2008. Pathological review of late cerebral radionecrosis. Brain tumor pathology 25, 51.

Round 2

Reviewer 2 Report

Responsiveness of the authors is appreciated. Revision improved the manuscript. A few remaining comments are:

  1. Authors should carefully revise the statements about Figure 1 as there seems to be discrepancy between what is meant and what is being said: “At the TCS low dose (10-8 M), TCS showed no effect on the number of primary dendrites compared to that in vehicle group (F2,150=16.69, p<0.0001) (Figure 1B). However, the number of secondary dendrites (F2,147=3.594, p=0.0300) in TCS 10-8 M group were no significant decreased compared to the vehicle group (Figure 1C).” Not effect, but p<0.0001?
  2. While antibody #9661 detects the cleaved caspase 3, the supplementary blot shows many bands, including bands above 30 kDa. While the correct band can be chosen to estimate the cleaved caspase 3 in immunoblotting, this cannot be done with immunofluorescence, indicating the signal may not reflect exclusively the cleaved caspase 3. This possibility (interpretation) should at least me mentioned in the discussion.
  3. Additional explanations about calculations are appreciated. However, it would be nice if percentage of apoptotic cells under control conditions is also reported (i.e. what was the fraction of apoptotic cells in controls).
  4. “In the present study, TCS reduced proliferation and promoted apoptosis of neuronal progenitor cells during brain development.” It is true that apoptosis was assessed in brains of experimental animals, but it was assessed in brain homogenates. So strictly speaking, apoptosis of neuronal progenitors during brain development was not directly estimated in vivo.
  5. Several typos remain.

Reviewer 3 Report

In the revised version the authors have addressed most of my concerns, included new data, and described their data and methods more comprehensively. This strengthens the conclusions and helps to better  understand their experimental strategy. The authors further provided detailed responses to my remarks, which together with the text changes clarified everything.

There are minor spelling mistakes that should be corrected:

eg:

Line 260:

2.6. TCS-exposure induces anxiety-like behavior, but DOES not affect depression-like behavior in mice 

line   145, last word should be: determined.

Author Response

We would like to thank the reviewer for the second suggestion to improve the quality of this manuscript. In the revised version, we have addressed the concerns that the reviewer raised.

  1. Line 260: 2.6. TCS-exposure induces anxiety-like behavior, but DOES not affect depression-like behavior in mice

Response: Again, we would like to thank you for your comment. We have changed this sentence by “TCS-exposure induces anxiety-like behavior, but does not affect depression-like behavior in mice”

  1. line 145, last word should be: determined.

Response: We have changed “determine” to “determined” at line 145.